# Trial-level sequence modeling reveals hidden dynamics of dual-task interference

Rick den Otter *, Anna Dame, Sjoerd Stuit, Leendert van Maanen

Helmholtz Institute, Department of Experimental Psychology, Utrecht University, Utrecht, The Netherlands

* r.denotter@uu.nl

## Abstract

Theories of dual-task interference assume that the same cognitive operations underlie multitasking regardless of stimulus timing, yet this core assumption has remained untested due to methodological limitations of behavioral averaging. Here, we combine hidden multivariate pattern (HMP) analysis with deep spatiotemporal sequence modeling of single-trial EEG to uncover the neural dynamics of multitasking in the psychological refractory period (PRP) paradigm. Using a deep spatiotemporal sequence model trained on Long stimulus-onset asynchrony (SOA) trials, we identify *Encoding*, *Central*, and *Response* operations and show that these same operations occur in the Short SOA condition, demonstrating shared cognitive processes across interference conditions. Additionally, trial-level decoding reveals multiple distinct sequences of cognitive operations across both tasks during interference, varying both within and across individuals. These sequences predict behavioral differences in reaction time and accuracy, revealing how interference timing within the cognitive operation sequence influences performance. In other words, we found trial-by-trial variability related to individual strategies directly affecting accuracy and reaction time (RT). Our findings challenge static bottleneck accounts and establish trial-level sequence modeling as a powerful tool to investigate the hidden dynamics of multitasking.

## Author summary

In our daily lives, we are often required to juggle multiple tasks at once, for example when operating an in-car device while driving. While multitasking is increasingly common, research shows that we often perform worse when multitasking, on average. Researchers have long been interested in when, how, and why this performance decline occurs, but most studies rely on averages that obscure how behavior changes at an individual trial level. In this paper, we use brain activity measured with electroencephalography (EEG) to identify recurring cognitive building blocks underlying simple decisions on individual trials. In the

**Data availability statement:** The data is made available openly at https://osf.io/5ub8z [34] by the original authors, the code is available at https://github.com/rickdott/prp.

**Funding:** The author(s) received no specific funding for this work.

**Competing interests:** The authors have declared that no competing interests exist.

task we use, participants first perform a visual decision, followed by an auditory decision. The delay between the first and the second task was manipulated. When the delay is short enough, the tasks overlap and multitasking is more common. We find that the building blocks for performing both tasks remain the same when multitasking, which until now has only been assumed, but not directly demonstrated. Additionally, we find that how the building blocks of both tasks are ordered and combined varies across trials, and this variation predicts differences in response speed and accuracy for the first task. These findings show that multitasking behavior is more dynamic than traditionally assumed.

## 1. Introduction

In today's fast-paced, information-rich world, we are often required to juggle multiple tasks at once. Because of this requirement, the ability to multitask effectively has become increasingly important for success in modern environments. Yet research consistently shows that multitasking comes at a cost, typically resulting in slower RTs and more errors [1,2].

An important theoretical notion of why multitasking costs extra time is that of a central bottleneck [3–5]. The central bottleneck theory assumes that individuals performing multiple tasks at once require more time because some cognitive processes are shared among the tasks, and cannot be executed in parallel. As a consequence, processing of a task may be delayed, leading to slower actions. This idea has traditionally been supported by the so-called psychological refractory period (PRP) paradigm [4,6–11]. In these experiments the time between sequential stimulus presentations, or stimulus-onset asynchrony (SOA), is manipulated to investigate the effect of dual-task interference on task performance. Distinct modalities such as visual and auditory are generally used to avoid perceptual interference, where both tasks are competing for the same sensory system. The RT of both tasks is then analyzed to theorize about what occurs during execution of the concurrent tasks. The typical finding is that the response time to the task that is presented first (Task 1) is not influenced by the stimulus onset of the task that is presented second (Task 2). The difference in Task 2 RT between the Long and Short SOA conditions is known as the PRP effect, a robust finding where shorter SOAs leads to longer Task 2 RTs [4,12,13]. The PRP effect has been the key piece of evidence for the central bottleneck theory [3,4], as it posits that a central, executive, process is required for both tasks. As this assumed process is only able to complete one task at a time, overlapping tasks yield slower responses for Task 2 [4].

However, alternative theories have been proposed to explain the PRP effect. In particular, the capacity sharing theory [14] states that central processing occurs in parallel, contrary to the seriality assumption of the central bottleneck theory. A limited amount of capacity is shared between both tasks, possibly restricted to central processes. An important prediction of capacity sharing – in contrast to the central bottleneck theory – is that the reaction times to Task 1 are also affected. Research

has tried to disentangle these theories by using experimental interventions that are meant to only affect Task 2 processes, for example varying the difficulty of the Task 2 stimulus [15], or increasing similarity between response types [16]. In these cases, if for example a more difficult Task 2 also affects Task 1 reaction times, that would support the capacity sharing theory. While these findings fit with the capacity sharing theory, the central bottleneck theory can account for these effects by distinguishing between response activation and response identification stages [13,17].

More recent theories have started to shift away from the focus on whether processing is parallel or serial [13,17], concluding that both are possible as a form of adaptive behavior. Instead, they focus on which contextual conditions motivate people to choose a serial or parallel execution. For example, the PRP effect can decrease with practice [18,19] (but does not disappear [20,21]), possibly indicating a shift from serial to parallel processing [22], although it can be argued that people simply become fast at each task separately and thus encounter costs of parallel execution less often [23]. However, there are known contextual variables that influence seriality of processing, such as priority instructions [24], expected temporal overlap [25], and task difficulty [26]. Additionally, individual differences possibly influence multitasking, as individuals differ in how frequently and efficiently they multitask [27,28]. These findings indicate that rather than being static, multitasking may be much more dynamic [5,29,30].

This dynamic perspective on behavior in the PRP paradigm is supported by neuro-imaging studies. A body of electro-encephalogram (EEG)/event-related potential (ERP) research has examined the neural dynamics of PRP dual-task performance. These studies typically track task-specific ERP components associated with perceptual processing (such as N2pc, posterior contralateral negativity (PCN)) and response preparation (lateralized readiness potential (LRP)). Consistent with the central bottleneck theory, several studies report that central processing affects ERP components related to perceptual [31–33] and motor [33] processing. However, a potential limitation of this work is that it relies on trial-averaged ERPs and thus assumes a fixed temporal structure across trials, which might not be the case. The timing of processing stages can vary from trial to trial, and averaging may obscure important variability in dual-task interference. In other modalities, [29] used magnetoencephalography (MEG) to show that Task 1 and Task 2 operations operate in parallel at first, but repel each other in the later part of the combined task. This means that Task 1 processes shorten, while Task 2 processes lengthen or are postponed. Their findings are incompatible with both the central bottleneck and capacity sharing theories, which is why the authors advocate for a new theoretical framework. Relatedly, [30] used ultrafast functional magnetic resonance imaging (fMRI) to draw conclusions about when serial processing does and does not occur, making the case for a nuanced central bottleneck theory.

Independent of the discussion between serial, parallel, or a more nuanced perspective, all theories on dual-task interference seem to assume – either explicitly or implicitly – that the same cognitive operations occur when tasks overlap (i.e., a Short SOA) as when they do not overlap (i.e., a Long SOA). For example, many theories refer to operations simply being shortened or delayed (e.g., [4,7,9,10,12,13,17]). [29] and [30] argue that task-related activity occurs at specific time-points across SOA conditions, implicitly assuming that the same cognitive operations occur in both conditions.

In the current paper, we will put this central assumption to the test. Specifically, we decompose EEG from a standard PRP task [34] into separate cognitive operations, using a machine learning (ML) method called hidden multivariate pattern (HMP) [35] analysis. Based on the EEG patterns and cognitive operation probabilities obtained this way from the Long SOA condition where tasks do not overlap, we next trained a deep spatiotemporal sequence model [36] to decompose the EEG at the single-trial level. This decomposition allows us to identify which cognitive operations occur on each trial, and compare their representations across SOA conditions to address whether these are comparable or not.

An additional benefit of our approach is that the single-trial decomposition reveals the specific sequence of cognitive operations on each trial. This way, we can understand whether these sequences are stable across trials, or whether there are large intra- and inter-individual differences in how the two tasks are combined when interference occurs. General theories of multitasking predict that such sequences can differ from trial to trial, depending on task contingencies [5], but this has never been identified in the PRP paradigm. In contrast, most PRP studies assume a certain homogeneity of the

sequences across trials, allowing for simple RT averaging [9,14,19]. Relatedly, [29] assumes that cognitive operations occur at the same time across trials and participants. This assumption allowed the training of ML classifiers at specific points in time, and testing at all other times to track cognitive operations. Similarly, [30] assumes that each trial unfolds in the same way, by averaging the hemodynamic response. Through our method of detecting cognitive operations at the trial level, we will be able to test these assumptions as well.

## 2. Methods

### 2.1. Ethics statement

This study re-used an openly available dataset [34]. The data was collected and shared in accordance with the ethical guidelines and approval of the original authors. No new data were collected, and therefore additional ethical approval was not required for the present work.

### 2.2. Experiment design

The experimental data used is from a previous study by R. Steinhauser and M. Steinhauser [34]. In this study's Experiment 1, 24 healthy participants (2 male) completed a three-choice flanker task (Task 1) followed by a two-choice pitch discrimination task (Task 2). The flanker task's squares could be three colors: red, yellow, or blue. Both flanker squares were always the same color, and this color was always different from the central, target square, meaning that target and flanker were never congruent, as would be typical in most flanker paradigms (e.g., [37]). The pitch discrimination task contained low (400 Hz) and high (900 Hz) sine tones. While performing the tasks, participants had to respond to the flanker task first by indicating the color of the target square, and to the pitch discrimination task second by indicating the pitch of the tone. Task 2 started after an SOA of 300 ms (the Short condition) or 1200 ms (the Long condition), chosen randomly within block. Participants used either the "Y", "X", "C" buttons, or the ",", ".", "-" buttons (QWERTZ layout) to respond to Task 1. In Task 2, participants used the up and down arrow keys or the "A" and "Y" buttons (QWERTZ layout). The color-to-key mapping was counterbalanced across participants and for one half of the participants the hand used was reversed. First, participants practiced during several blocks, ensuring that the experimental paradigm and color-to-key mapping were sufficiently learned. Finally, 10 blocks of 108 trials each were conducted. See [34] for more details about the experimental paradigm.

### 2.3. Data collection and preprocessing

The EEG data were collected using 64 electrodes placed according to the 10–20 positioning system at a sampling rate of 512 Hz using a BioSemi Active-Two system (BioSemi Instrumentation, Amsterdam, The Netherlands). Using MNE-Python [38], we first band-pass filtered (1–100 Hz) with a 50 Hz notch filter to suppress line noise. individual component analysis (ICA) [39] is applied after downsampling to 200 Hz to identify and remove ocular and other stereotyped artifacts. Removal of the ICA components is performed on the original, non-downsampled data. Long and Short interval condition triggers were reconstructed based on the interval between stimulus onsets. We created two epochs out of each trial, from -0.25 s before to 2.0 s after stimulus onset for each of the two tasks. We linearly de-trended the epochs using *mne*'s [38] implementation and baseline-corrected both Task 1 and 2 epochs using pre-Task 1 stimulus information, to ensure that no task-relevant activity was used for baseline correction. We used *autoreject* [40] to remove bad trials (Task 1 M: 64.37, SD: 75.62; Task 2 M: 58.50, SD: 55.26). Finally, we split the data from the Long condition into 85% and 15% training and validation sets, by participant.

### 2.4. Hidden multivariate pattern analysis

We used HMP [35] to estimate probability distributions for the onset of each cognitive operation. HMP assumes that the onsets of cognitive operations are represented by a significant event that is detectable as a change in the multivariate

EEG signal [35,41]. It estimates the most likely location of these events by jointly modeling the contribution of each channel to this change in signal and the time interval distribution between events. HMP has been successfully applied in many domains to disentangle cognitive operations [35,41–45]. In the current application, we fit HMP to the Long condition only. This is necessary because HMP relies on the temporal dependence of subsequent events [35]. In the Short SOA condition, this temporal dependence may not be present due to Task 2 operations occurring while Task 1 is being performed. With a Long SOA of 1200 ms, Task 1 and 2 overlap is negligible, as the 95th percentile of Task 1 RT is at 840 ms, which is substantially shorter than the Long SOA. Application of HMP to the Long SOA condition allows us to estimate the onset of the cognitive operations of each task reliably and independently. We cut off all epochs at RT, removing trials with an RT of lower than 0.2 s. The data were re-referenced to the average over all electrodes, and band-pass filtered between 1 and 50 Hz. Additionally, the 50 ms after RT is used by HMP, making the final response operation easier to estimate (cf. [35]). We perform principal component analysis (PCA) on the variance-covariance matrix averaged over all the data included in the training set, keeping 10 components. Using these components, we then fit HMP on Task 1 and 2 separately (Long SOA only), using PCA weights calculated over EEG data from participants in the training set only. Both HMP fits used the cumulative fitting procedure. We tuned the event width parameter by testing a range of different event widths (20–60 ms, steps of 5 ms, cf. [35]). The event width fits ranged from six Task 1 and six Task 2 events (when the event width was 20 ms), to three Task 1 and two Task 2 events (when the event width was 60 ms). An event width of 50 ms – giving three events – was determined as the most stable value by investigating the number of estimated events, meaning that this value and the surrounding event widths (45 ms and 55 ms) all resulted in three events in both tasks. To aid interpretation, we label the three estimated events according to dual-task interference theory [4]: *Encoding*, *Central*, and *Response*.

## 2.5. Model architecture

Our deep spatiotemporal sequence modeling approach (Fig 1) is based on the Mamba [36] state space model (SSM) architecture. Similar to our previous work [46], we first perform spatial feature extraction using a $1 \times 1 \times C$ point-wise convolution kernel, where $C$ is the number of EEG channels. Temporal dropout is then applied to reduce the model's reliance on specific time-points [47]. Temporal features are extracted at two different time scales, 12 ms and 36 ms, and the resulting features are concatenated along the feature dimension. Since trials are variable in length, we add a relative positional encoding vector as an additional feature. This vector is set to 0 until stimulus onset, increases linearly from 0 to 1 between stimulus onset and RT, and remains at 1 thereafter. This encourages the model to focus on relative rather than absolute time within the trial. The combined features are passed through five sequential Mamba [36] layers, integrating both spatial and temporal features. At this point, we depart from our previous work by including task-specific classification heads. These task-specific classification heads are used at inference time, allowing the model to learn shared representations across tasks while maintaining task-specific decision boundaries [48].

### 2.5.1. Model training.
We used the data from the Long condition to train the sequence model in a similar way to our previous work [46]. Since we validated different model hyperparameters in that work, we re-used these. We create a training and a validation set, normalizing each epoch by applying median average deviation $z$ scoring using information from the training set only. Additionally, we add a *Negative* class to the class labels, ensuring that at each time step, the sum probability is 1.0. While training, each epoch is jittered randomly, meaning that a new start and end point are chosen from additional samples that we kept before stimulus (250 ms) and after RT (300 ms). This is done to ensure that the sequence model does not learn to rely on specific points in time. Additionally, including samples that are only labeled as the *Negative* class gives the model more knowledge about what is specifically *not* the onset of a cognitive operation. We train a model on Task 1 and Task 2 at the same time, using a shared encoder but separate classification heads. We label each operation separately from the operations in the other task [48]. We also tried training the model on trials that started at stimulus onset of Task 1 and ended at RT of Task 2, but this attempt gave suboptimal results: While the model did find

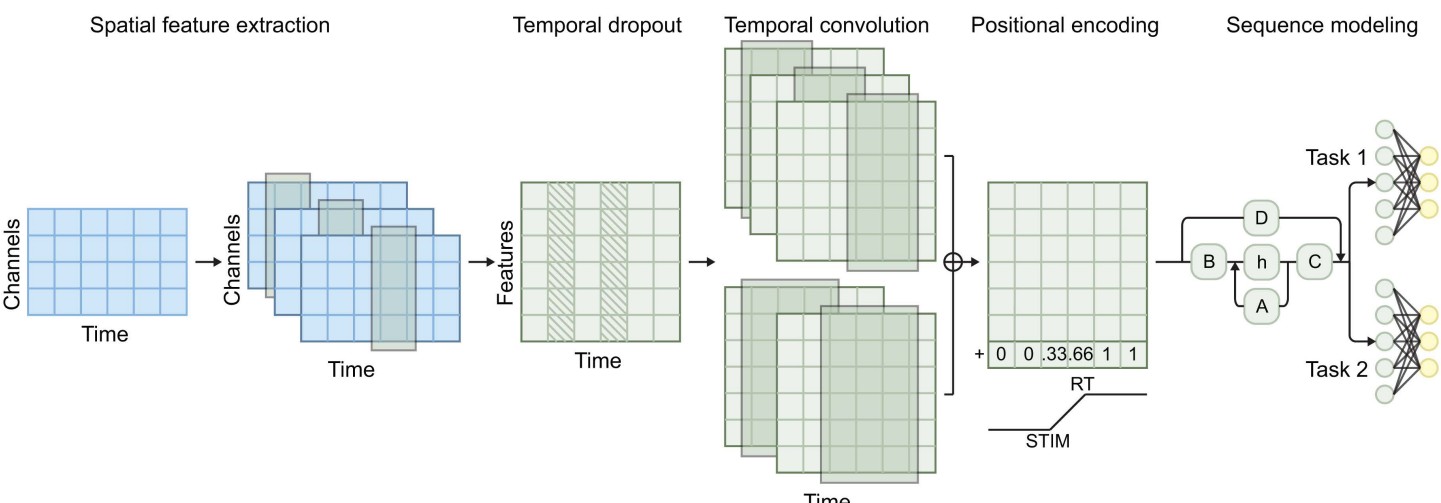

**Fig 1. Model architecture of the sequence model that predicts cognitive operations over time.** The model architecture used, blue indicates data, green indicates processing, yellow indicates output. First, spatial features are extracted from raw data, followed by temporal dropout. Then, temporal convolution is used to model temporal relationships at multiple time scales, after which positional encoding is added to the features. Finally, all features are fed into a Mamba sequence model followed by a task-specific fully connected layer.

cognitive operations, this model predicted implausible within-task orders of cognitive operations more often than the model trained on both tasks separately.

The model is trained using Kullback-Leibler (KL) divergence between HMP-estimated probabilities and predicted probabilities as the loss function. We use the NAdam [49] optimizer with a learning rate of 0.00005. We stopped training early if validation loss did not decrease for three subsequent epochs. We saved the model weights of the epoch with the lowest validation loss. We trained the model on an NVIDIA A40 GPU with 12 GB VRAM available. All code written for this paper is available through https://github.com/rickdott/prp.

### 2.6. Embedding analysis

Instead of separately analyzing the peak time of predicted cognitive events and the EEG at that point, we examined embeddings to understand the similarity between cognitive operations in the Short and Long conditions. Embeddings are the sequence model's learned transformation of the EEG, combining all relevant spatial and temporal information in one shared feature space [50]. To extract the embeddings for each cognitive operation, we considered the model, trained on all trials in the Long condition for 85% of participants. We then used this model to infer embeddings over all remaining trials in the Long condition, along with all trials for every participant in the Short condition. Embeddings are taken from the model after the Mamba [36] layers. At this point, the model's representation of the trial is a *#samples × features* tensor. We extract both this tensor and the predicted probabilities for each class. For each trial and for each cognitive operation the average embedding of 13 samples (52 ms) centered around the peak prediction was computed, closely corresponding to the event width used as HMP parameter.

For visualization purposes, we projected the embeddings to two dimensions using T-SNE [51] (scikit-learn 1.6.0, n_components = 2, random_state = 42, remaining parameters left at default values). This projection revealed several compact sub-clusters within the *Response* (R) cognitive operation; these sub-clusters were further analyzed in S1 Appendix.

## 2.7. Sequence analysis

To decode the sequence of cognitive operations across both tasks, we used our model, trained on HMP fits for Task 1 and Task 2. We applied the model to the testing data, resulting in task-specific probability distributions indicating the onset of each cognitive operation. To reconstruct the combined temporal structure of both tasks, we temporally aligned the Task 2 predictions by shifting them according to the SOA. Finally, rather than assigning a single onset time to each cognitive operation, we preserved the uncertainty in these estimates. We derived a posterior distribution over sequences for each trial by repeatedly sampling operation onset times (1000 samples per trial) from the full operation-specific probability distributions, weighted by their predicted probabilities. We retained all sampled sequences that satisfied within-task ordering constraints to ensure physiologically plausible ordering. To carry this uncertainty forward into our behavioral analyses, we used a multiple-imputation framework. We repeatedly sampled complete datasets from the trial-level sequence posteriors (100 imputations), refit the mixed-effects models for each dataset, and pooled parameter estimates using Rubin's rules [52].

## 3. Results

### 3.1. Determining the number and nature of cognitive operations

Performing HMP analysis on our data gave us a number of cognitive operations and their estimated position within each trial. The HMP fit on data from the Long condition resulted in three events in both Task 1 and Task 2, visualized in Fig 2. When interpreted as ERPs, we can view the events of Task 1 as a posterior N1 [53], related to visual processing. The second event could be a fronto-central N2, a classical conflict-monitoring signal often seen in flanker tasks [54]. The last event is a clear LRP pattern, associated with response selection and execution [50]. For Task 2, we observe a slightly clearer pattern resembling the N1-P2 complex, associated with auditory processing [54]. In this task, the sensory processing event is followed by what seems like a centro-parietal P3 [54,55]. Similar to Task 1, we see a clear LRP response-related event at the end of the trial. For easier discussion of these events, we label them according to the dual-task interference literature [4]: *Encoding* (E), *Central* (C), and *Response* (R).

### 3.2 Cross-decoding cognitive operations across condition

To address our main hypothesis that operations in the Long condition are also found in the Short condition, we applied a sequence model trained on Long SOA data to Short SOA data, as follows. For every trial, we extracted the sequence

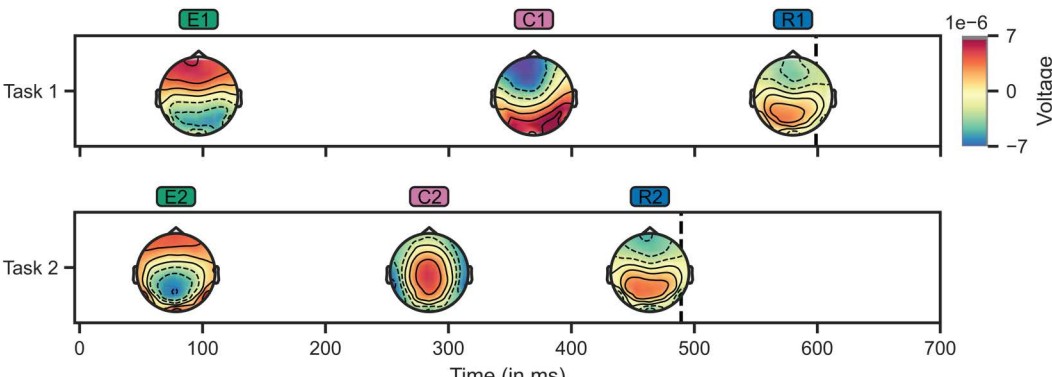

**Fig 2. Hidden Multivariate Pattern (HMP) analysis reveals three cognitive operations in both tasks.** Results of HMP, fit on data from the Long condition. Operation topographies are aligned to the most likely point in time at which each activation peak occurs. EEG activity at these points is averaged over all trials and participants to create the topographies. The vertical dashed lines denote the average RT. E: *Encoding*, C: *Central*, R: *Response*.

model's learned representation for each cognitive operation. We compared embeddings across conditions. If the EEG data is embedded similarly, then we argue that the same operations are found. In Fig 3a, we show that within each task, embeddings form three robust clusters corresponding to the *Encoding* (E), *Central* (C), and *Response* (R) cognitive operations. For every operation, the Short and Long SOA conditions occupy a similar region. The *Response* operation embeddings cluster into multiple sub-clusters (Fig 3a), we analyzed these clusters to find whether it was an aspect of the task, behavior, or the model that caused the clustering (See S1 Appendix). We found that the most likely explanation of these clusters was a response time binning, in the sense that the model represents the duration between the response event and the RT to optimize performance.

To quantify the observation from Fig 6a, we computed the representational similarity between all cognitive operations across both conditions, separately for each task. This addresses the question whether indeed a specific cognitive operation in the Short condition is more similar to that same cognitive operation in the Long condition than to any other cognitive operation in the Long condition. To this end, we performed 1000 resampling iterations. In each iteration, we randomly sampled (with replacement) 1000 samples from each of three different sets: The Short condition embeddings for the same cognitive operation (Task 1, total number of events: $n = 11985$, Task 2: $n = 12003$), the Long condition embeddings for the same cognitive operation (Task 1: $n = 2193$, Task 2: $n = 2194$), and the Long condition embeddings for other cognitive operations (Task 1: $n = 4386$, Task 2: $n = 4390$). We computed the absolute average pairwise cosine distance difference between samples from the Short/Same Task and the Long/Same Task and Long/Other Task sets. Across conditions, within task, the same-operation differences were distributed at essentially zero (median $= 0.00$, 95% *CI* [0.00, 0.02]), suggesting a high degree of similarity. The cosine distance to other operations, within task, was larger (median $= 0.50$, 95% *CI* [0.26, 0.54]), as demonstrated by the non-overlapping confidence intervals. These results show separation in cosine similarity between sets, indicating no meaningful difference between operations of the same task across delay conditions. Together,

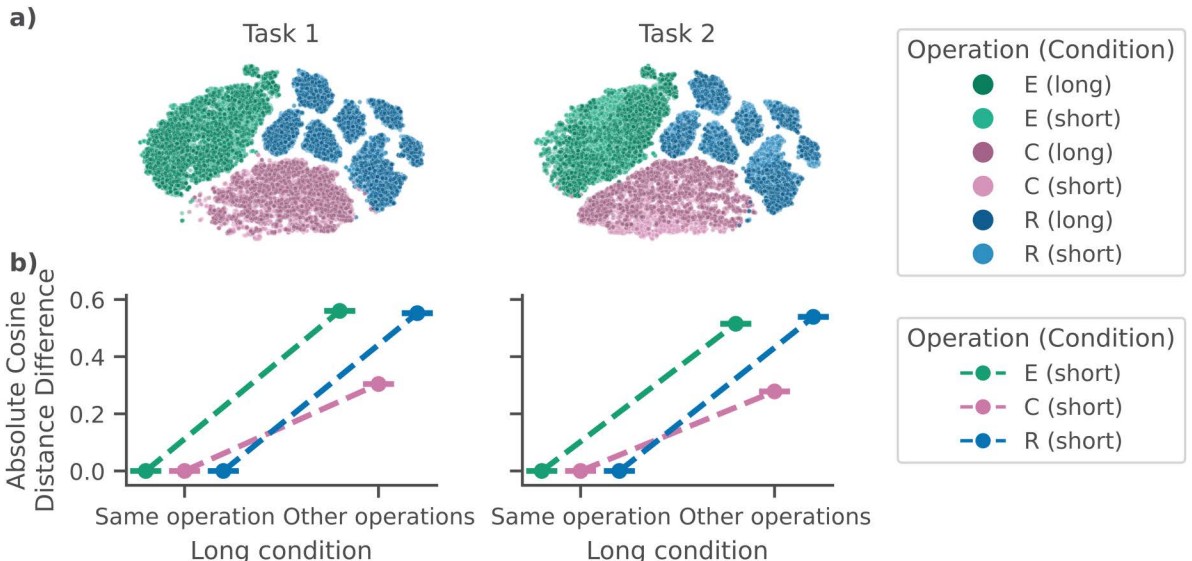

**Fig 3. Embeddings of cognitive operations cluster together across conditions. a)** Our model finds the same cognitive operations in the Short (unseen) condition as in the Long condition (only participants in the validation set used). In a 2D T-SNE [51] projection, each operation is clustered separately and the Long and Short condition's projections overlap, for both Task 1 and Task 2. **b)** In high-dimensional embedding space, the absolute cosine distance difference (Y-axis) between embeddings of cognitive operations in the Short condition and those *same* operations in the same task (X-axis) in the Long condition is negligible (95% *CI* includes 0), and smaller than the difference between operations in the Short condition, and the *other* operations in the same task (X-axis) in the Long condition. Plotted points are the median, error bars represent 95% CI. E: *Encoding*; C: *Central*; R: *Response*.

PLOS Computational Biology

the results indicate that the model embeds Short cognitive operations in the same regions of the learned high-dimensional embedding space as their Long counterparts, demonstrating that the same *Encoding*, *Central*, and *Response* operations are present in both SOA conditions.

### 3.3. Decoding the sequence of cognitive operations across tasks

Having demonstrated that we find identical cognitive operations for both the Short and the Long conditions, we can now address our secondary research question and explore how the timing of cognitive operations across tasks affects behavior. We reconstructed Task 1 and Task 2 trials back to their original temporal structure, and combined all class probabilities. This allows us to inspect the sequence of cognitive operations across both tasks. We found that a variety of sequences is used. In the Long condition, as expected, all cognitive operations of Task 1 are initiated before Task 2. Within task, we find the expected order of E, C, and R for both tasks. Interestingly, in the Short condition, this sequence also occurs, along with five others (Fig 4a). Here, we excluded invalid sequences (e.g., $R_1 \rightarrow E_1 \rightarrow C_1 \rightarrow C_2 \rightarrow E_2 \rightarrow R_2$), which occurred only marginally (0.47% of trials, see Table 1 for sequence definitions). In five of the six sequences (B-F), cognitive operations from the second task occurred prior to the completion of the first task. We call these the Short SOA-specific sequences.

Task 1 behavior is affected by sequence type (Fig 4b and c). To quantify this observation, we applied linear mixed effects models to RT and accuracy.

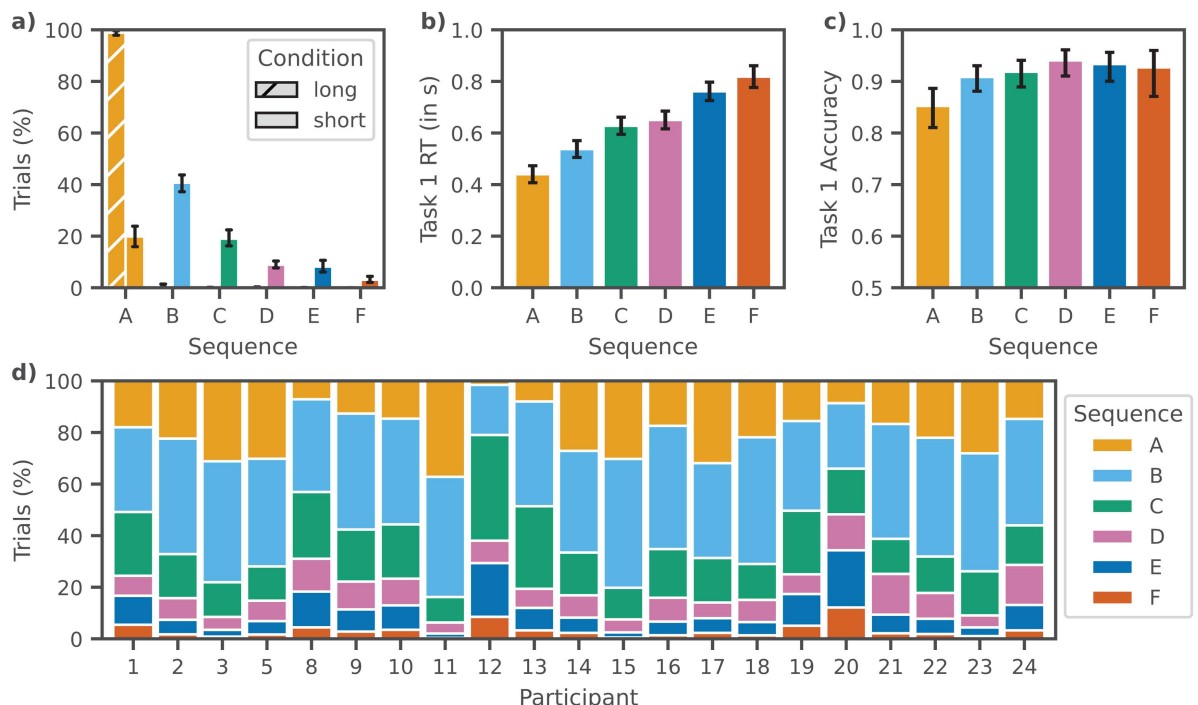

**Fig 4. Different sequences of cognitive operations across tasks occur and predict Task 1 behavior. a)** Proportion of sequence occurrences per condition, averaged across participants. In the Long condition, a single sequence was used predominantly. In the Short condition, multiple different sequences were used (see Table 1). Error bars indicate 95% confidence intervals across participants. **b/c)** The sequences used differ significantly in Task 1 RT and accuracy, indicating that the timing of interference affects task performance. Shown values indicate pooled estimates and 95% confidence intervals from fitted linear mixed-effects models. **d)** Participants in the Short condition employ a variety of sequences, and the distribution differs substantially over participants.

**Table 1. Mapping of sequence labels to the corresponding sequences of cognitive operations.** $E_1$ is *Encoding* for Task 1, $C_1$ is *Central* for Task 1, $R_1$ is *Response* for Task 1. The same convention was followed for Task 2.

| Sequence label | Sequence of cognitive operations |
| --- | --- |
| A | $E_1 \rightarrow C_1 \rightarrow R_1 \rightarrow E_2 \rightarrow C_2 \rightarrow R_2$ |
| B | $E_1 \rightarrow C_1 \rightarrow E_2 \rightarrow R_1 \rightarrow C_2 \rightarrow R_2$ |
| C | $E_1 \rightarrow E_2 \rightarrow C_1 \rightarrow R_1 \rightarrow C_2 \rightarrow R_2$ |
| D | $E_1 \rightarrow C_1 \rightarrow E_2 \rightarrow C_2 \rightarrow R_1 \rightarrow R_2$ |
| E | $E_1 \rightarrow E_2 \rightarrow C_1 \rightarrow C_2 \rightarrow R_1 \rightarrow R_2$ |
| F | $E_1 \rightarrow E_2 \rightarrow C_2 \rightarrow C_1 \rightarrow R_1 \rightarrow R_2$ |

For Task 1 RT, the pooled linear mixed-effects models ($RT_{T1} \sim$ sequence + (1|participant)) showed a robust main effect of sequence, ($F(5, 1415.54) = 448.91$, $p < .001$). Holm-adjusted pairwise contrasts indicated that most sequence pairs differed significantly ($p_{adj} < .001$); the difference between sequences C and D was small but reliable ($p_{adj} = .004$), and the difference between sequences E and F was modest ($p_{adj} = .001$). For Task 1 accuracy, the logistic mixed-effects model indicated a main effect of sequence ($F(5, 2204.65) = 11.65$, $p < .001$). Pairwise comparisons showed that accuracy differences were primarily driven by contrasts involving sequence A; most other sequence pairs did not differ reliably after Holm correction (all $p_{adj} > .05$). These results suggests that the point in the combined sequence of cognitive operations where Task 2 interferes with Task 1 influences the efficiency with which the participants execute Task 1, resulting in significant differences in RT, and sometimes accuracy. Note that we observe behavioral variability across sequences even when the mean Task 1 RT across conditions is very similar (Short: $M = 567$ ms ($SE = 1.79$ ms); Long: $M = 571$ ms ($SE = 4.06$ ms)).

Even though Task 2 interference is dependent on Task 1 performance and Task 2 behavior is not relevant to our research question, we also checked whether we would find behavioral differences in Task 2 performance, as expected based on classical PRP results [4]. We found that there was a main effect of sequence type on Task 2 RT, ($F(5, 2022.67) = 71.34$, $p < .001$). Pairwise comparisons (Holm-adjusted) indicated significant differences between most sequence pairs ($p_{adj} < .001$); except for sequences B and D ($p_{adj} = .009$), and sequences E and F ($p_{adj} = .01$). For Task 2 accuracy, we found no main effect for sequence ($F(5, 2306.44) = 0.49$, $p = 0.783$) and refrained from further analyzing it.

Additionally, we examined variation in sequence use at the participant level. As shown in Fig 4d, there was substantial individual variability in the frequency with which different sequences were used. Some participants preferred sequences in which all Task 1 operations preceded all Task 2 operations, while others more frequently used sequences where Task 2 processing started before the end of Task 1 processing. The observed heterogeneity suggests that participants differ in their approach to dual-task processing. Although not the primary focus of our study, these differences highlight how individual cognitive strategies might shape how interference is managed.

## 4. Discussion

In this study, we aimed to investigate cognitive operations in multitasking at a neural level. To do so, we used HMP-derived estimations of the onset of cognitive operations and deep spatiotemporal sequence model analyses. We were able to decode cognitive operation sequences at a resolution previously unattainable with behavioral or imaging data. Firstly, we found that three cognitive operations occurred in both tasks. Secondly, and in accordance with our main hypothesis, we determined that the same cognitive operations occurred independent of SOA condition. Additionally, we were able to decode trial-level sequences of cognitive operations across both tasks. We identified two main multitasking strategies. Here, a strategy refers to repeatable patterns of temporal organization, without implying conscious or deliberate control. In our analysis, this repeatable pattern is a sequence of cognitive operations. We found a non-overlapping strategy, where tasks are processed one after the other just like in single-task conditions, and overlapping strategies,

where processing of the second task begins before the first task is completed. Classical theories [14,56] suggest either no, or a limited effect of dual-task interference on Task 1 RT. However, we find an effect on Task 1 RT that depends on the specific strategy used. We interpret this as evidence that Task 1 RTs are not generated by a homogeneous process. Note that our model nearly always predicts one sequence (Sequence A) in the Long condition, corresponding to a non-overlapping sequence of cognitive operations. This serves as a validation of our method since the model does not detect overlapping sequences when the tasks are temporally separated. By explaining interference at the single-trial level, we clarify so-far inconclusive findings on the effect of interference on RT to Task 1.

Our findings provide new insights into dual-task interference that challenge classical [4,14] and modern [29,30] accounts. Contrary to the central bottleneck theory, which assumes that Task 1 processing is not affected by interference, our results show that Task 2 can influence Task 1, depending on the strategy employed. The capacity sharing theory could in principle account for our current findings by assuming that the distribution of cognitive resources between the two tasks varies across trials. However, the capacity sharing theory offers no way of determining which trials reflect which distribution of cognitive resources. In other words, these models make no prediction on the trial-level dynamics of dual-task interference. To summarize, our results imply that models of dual-task interference must account for dynamic, trial-level fluctuations in strategy use instead of assuming a fixed processing architecture, as our results show that this fixed architecture does not exist. The issue of assuming a fixed architecture is also prevalent in modern decoding approaches, where research relies on temporal averaging across trials, while our method operates at the single-trial level. This allows us to capture trial-by-trial variability in cognitive operation sequences that would otherwise be obscured by aggregating, and thus assuming a fixed architecture. Trial-by-trial variability reflects more than just noise, it reveals systematic differences in how participants approach multitasking. Participants do not rely on a single strategy of multitasking, but instead they flexibly switch between strategies. We observe that the non-overlapping strategy is faster but less accurate, whereas the different overlapping strategies take more time but result in higher accuracy. Interestingly, despite the presence of overlapping versus non-overlapping strategies, mean Task 1 RTs were highly similar. Averaging over all trials would thus suggest little difference between conditions. However, our trial-level method reveals underlying variability, where participants employ distinct strategies, resulting in substantial behavioral differences. Together, these results suggest that classical theories alone cannot fully explain the observed variability in multitasking behavior. To fully understand interference in multitasking, we must consider more than a condition-averaged view of behavior, and embrace trial-level dynamics and individual differences in cognitive strategy use. These individual differences in strategy use may reflect differences in cognitive control or executive function [28,57,58]. Individuals with higher working memory capacity or greater cognitive flexibility might prefer parallel processing [59], while less flexible individuals may default to sequential strategies.

Although investigating multitasking behavior at single-trial level underscores the importance of individual variability in research, this also raises questions about limitations of the current method for capturing and interpreting such variability. Firstly, our ground truth for when the onsets of cognitive operations occur consists of HMP-derived event onset probability distributions. The signal-to-noise ratio of EEG makes it difficult to precisely determine when a cognitive operation starts. HMP-derived probabilities are estimates of when cognitive operations occur, rather than an objective ground truth. This could limit the performance of our model as most optimization algorithms assume that labels are accurate, while this may not always be the case in our data. However, we expect that this affects all cognitive operations equally, entailing that the model would still capture the overall pattern of cognitive operations, and thus does not invalidate our conclusions. Another consequence of the uncertainty in ground truth that has a different effect is that since the true timing of cognitive operations is uncertain, the model's performance metrics should be interpreted as relative rather than absolute. In other words, the model optimizes against approximations of the truth rather than known cognitive operation onsets. To partially address this limitation, we use the KL divergence as our loss function, which helps to account for uncertainty in class labels without over-penalizing plausible alternative predictions [60]. An additional limitation regards a data-driven hypothesis that is not verifiable with our data, namely: We hypothesize that the proportion of default, non-overlapping, strategy use is the main

driver of the Task 2 interference effect on Task 1 RT. This implies a gradual effect, which we unfortunately cannot test in the current data set. In the original study only two SOA conditions were included, 300 ms and 1200 ms. This suffices to show a PRP effect, but does not allow us to show that the proportion of default non-overlapping strategy use is what drives Task 2 interference effect on Task 1 RT. More SOA values would allow us to show a slope of proportion, where a higher proportion would imply a lower average Task 1 RT. Lastly, our methodological approach requires the Long condition to have a sufficiently large SOA value, combined with a sufficiently quick Task 1, to ensure that there is no task overlap on sufficient trials. However, experiments deploying the PRP paradigm often do not include a long enough SOA [32,33,61] to ensure the absence of task overlap. This limits the cross-dataset generalizability of our approach.

Nevertheless, being able to investigate trial-level differences in multitasking opens up many venues of research. While our and other research [13,17,29,30] has shown that both serial and parallel execution of multiple tasks are possible at task level, our method might be used to investigate whether, at a cognitive operation level, execution is serial or parallel. While HMP enforces a serial structure, through fitting on both conditions separately we do not enforce this across tasks. By looking at the time intervals between cognitive operations when operations overlap versus when there is no overlap we may be able to find that specific cognitive operations affect this interval differently. If we find that the time interval between a cognitive operation and its successor is not modulated by the presence of an event of the other task occurring between them, then this would show that there is at least partial parallel execution at the cognitive operation level. As these findings do not relate to our main hypotheses, we did not investigate them further. Relatedly, we observe that the sequences appear to introduce behavioral trade-offs, as slower sequences are generally more accurate. This pattern of behavior suggests a speed-accuracy trade-off (e.g., [62–65]). However, this cannot be the whole story, as we observe no difference in accuracy between sequence pairs that do differ in RT. This is especially clear in sequence pairs B and C, and D and E. To understand the behavioral relationships between speed and accuracy, one could deploy evidence accumulation modeling, making use of the full RT-accuracy distribution. [66,67]. This could be applied either on the observed RTs or the inter-event intervals that follow from our analyses (cf. [42,45]).

Our sequence modeling approach provides a framework for studying trial-level cognitive processes that traditional averaging-based analyses cannot capture. By decoding the temporal structure of cognitive operations on a trial-by-trial level, this method allows researchers to investigate when and how different strategies overlap in time. This framework could be applied to other experimental paradigms in cognitive neuroscience where temporally overlapping cognitive operations play a role. For example, in the attentional blink paradigm [68,69] participants are asked to report two target stimuli in a fast-paced stream of distractor stimuli. Trial-by-trial variability in the timing between the encoding and central operations of the first and second target stimulus might cause impaired detection of the second target (cf. [70]). Another interesting direction may be to investigate the contextual factors influencing non-overlapping versus overlapping strategy use. Recent research [17] has shown that instructions and task-specific factors influence the ratio between serial and parallel execution. Our method could be used to identify how specific contextual manipulations bias the prevalence of overlapping versus non-overlapping strategies. Lastly, since this method is able to decode strategy use at trial level, it could be used to investigate which individual differences influence non-overlapping and overlapping strategy use [13,28,57–59]. In doing so, this method could help uncover how stable cognitive and personality traits influence multitasking strategy, opening pathways for more personalized models of cognition.

## 5. Conclusion

In this study, we set out to investigate two aspects of dual-task interference: whether cognitive operations remained the same across different SOA conditions, and how dual-task interference affects Task 1 performance. We applied hidden multivariate pattern (HMP) analysis to detect three distinct cognitive operations in each task, *Encoding*, *Central*, and *Response*. We used deep spatiotemporal sequence modeling to find that whether dual-task interference occurs or not, the cognitive operations remain the same, in agreement with the central – previously untested – assumption underlying

all PRP research. Finally, we found that the combined sequence of cognitive operations across task differs per trial when interference occurs. Which sequence is used substantially affects participants' reaction time (RT) and performance in Task 1. The ability to classify trials to a set of different sequences allows for a more fine-grained analysis of dual-task interference, alleviating the need for condition-level averaging.

## Supporting information

**S1 Appendix. Response operation cluster analysis.** Contains analysis of the clusters found in the projection of response operation embeddings.
(PDF)

## Author contributions

**Conceptualization:** Rick den Otter, Sjoerd Stuit, Leendert van Maanen.

**Formal analysis:** Rick den Otter.

**Methodology:** Rick den Otter, Anna Dame.

**Software:** Rick den Otter.

**Supervision:** Sjoerd Stuit, Leendert van Maanen.

**Validation:** Rick den Otter.

**Visualization:** Rick den Otter.

**Writing – original draft:** Rick den Otter.

**Writing – review & editing:** Rick den Otter, Anna Dame, Sjoerd Stuit, Leendert van Maanen.

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
