## [Decision Letter · Decision Letter 0]

30 Jan 2026

PCOMPBIOL-D-25-02374

Trial-level sequence modeling reveals hidden dynamics of dual-task interference

PLOS Computational Biology

Dear Dr. den Otter,

Thank you for submitting your manuscript to PLOS Computational Biology. After careful consideration, we feel that it has merit but does not fully meet PLOS Computational Biology's publication criteria as it currently stands. Therefore, we invite you to submit a revised version of the manuscript that addresses the points raised during the review process.

We look forward to receiving your revised manuscript.

Kind regards,

Boris S. Gutkin

Academic Editor

PLOS Computational Biology

Andrea E. Martin

Section Editor

PLOS Computational Biology

**Journal Requirements:**

4) We notice that your supplementary information is included in the manuscript file. Please remove them and upload them with the file type 'Supporting Information'. Please ensure that each Supporting Information file has a legend listed in the manuscript after the references list.

**Reviewers' comments:**

Reviewer's Responses to Questions

**Comments to the Authors:**

Reviewer #1: The paper uses EEG data to analyze two issues of relevance to studies of the psychological refractory period (PRP). The data involve a short SOA condition where Task 2 begins just 300 msec after Task 1 and Task 1 typically has not been completed and a long SOA condition where Task 2 begins 1000 msec after Task 1 and Task 1 typically has been completed. The two questions are (1) whether the stages of the two tasks remain the same in the two conditions and (2) whether it is really true that performance of Task 1 is not changed in the short SOA condition.

The answer to the first question is not compelling, as detailed below. The answer to the second question has a potential confusion of causal direction, also as detailed below.

The paper uses hidden multivariate pattern (HMP) analysis to identify the stages in the task. The analysis focuses on the EEG data from the onset of the task to 50 msec after the task response and identifies 50 msec event patterns in the data that mark stages in processing. Four analyses are performed – of both tasks at both SOAs. The paper reports three event patterns in all 4 analysis which are labeled as Encoding, Central, and Response. A more thorough reporting of the 4 analyses is needed. We are only shown the Figure 2 summary of the two tasks in the long condition. There is no evidence that there are these three event patterns, not more or less. Such confirmation of number of event patterns is common in HMP analysis.

The first question is whether the event patterns are the same for the two SOAs. One might have expected a standard statistical comparison of electrode values or PCA values associated with the event patterns. However, a deep learning architecture is used to extract embeddings associated with the event patterns and examine the similarity between long and short embeddings. It is not clear (to me) in detail what the embeddings mean. Figure 3 does not really help. What does “Other” in part B refer to? The caption refers to “other operations” – are these other operations in the same task or the other task. To the extent that there is a statistical conclusion it is that same distances are smaller than “other” distances. However, one would want to know whether the differences between the same operations in short and long is statistically different from 0.

The second question is whether the first task is unchanged in the short condition. To do this separate HPM analyses are performed of the overlapping EEG periods for the two tasks. In the short SOA condition, one might have expected an analysis of the single period from the beginning of the first task to the end of the second task, looking for event patterns from the two tasks. The paper’s approach of separate HPM analyses offers some advantages, but one has to worry about a event pattern from the one task being attributed to the other.

The HMP gives the most probable location of the event pattern on each trial, and I assume this is what produces the sequences for Table 1 and Figure 4. The sequences are the orders of those most probable locations. Given the nature of the HPM and the fact that task 1 finishes before task 2 starts in the long condition there is only one possible ordering of conditions: E1 → C1 → R1 → E2 → C2 → R2. However, there are 6 sequences in the short condition counting the ignored: E1 → E2 → C2 → C1 → R1 → R2. The duration and accuracy of Task 1 varies with the sequence. The different sequence are interpreted as different strategies employed by the participant. However, the sequences can be viewed as just where the stages of one tasks fell compared to the independent stages of the second task. There is something of a speed-accuracy trade-off in Task 1. The only cases where one could get the E1 → C1 → R1 → E2 → C2 → R2 sequence in the Short SOA is when task 1 completes near the beginning of when task 2 is presented (300 msec. later). It is a necessity that this sequence is associated with brief response time and responding so rapidly seems the reason for the low accuracy.

It is a mistake to treat these sequences as causing behavioral differences and ignore the likely possibility that it is differences in the speed of performance of the two tasks that are producing the different sequences. Also, as the authors admit, it is deeply problematic to take the maximum likelihood locations as reflecting when the operations are performed. The locations are far from certain and presumably there are many cases where the order of two event patterns could have been flipped without much cost to likelihood.

Reviewer #2: The present study investigated the processing architecture and processing strategy of task performance in the PRP paradigm. While previous studies applied RT data, imaging data, EEG data, etc. in an average way, the present study investigated PRP task performance on a trial level. In particular, the processing stages of encoding, central, and response execution were separated, and the processing sequence of these stages was analysed in a re-analysis of an existing data set of Steinhauser and Steinhauser. The authors found individual trial-by-trial variability related to strategies that directly affect accuracy and RTs. The study discusses these findings in the context of tradition and more modern theories on multitasking.

I like the methodological approach of the manuscript and the conclusions that can be derived from the HMP method. However, before recommending this manuscript for publication, there are some substantial comments to solve. These comments are rather from a theoretical than from a methodological perspective and listed in the order of the structure of the manuscript.

Line 20: “In contrast, Task 2 responses are typically delayed.” This sentence is not correct in itself in the context of PRP dual tasks. It is the long versus the short Task 2 condition which is delayed, and which is correctly stated in the following sentence. Therefore, I suggest cutting this sentence (“In contrast, Task 2 …”).

Lines 46 – 48: “For example, the PRP effect can decrease with practice [18, 19], possibly indicating a shift from serial to parallel processing [20], although it can be argued that people simply become fast at each task separately and thus encounter costs of parallel execution less often [21].” Please, refer to more recent research on practice effects such as Strobach & Schubert (2017: Strobach, T., & Schubert, T. (2017). No evidence for task automatization after dual-task training in younger and older adults. Psychology and Aging, 32(1), 28-41.) or Schubert et al. (in press: Schubert, T., Liepelt, R., & Strobach, T. (in press). Evidence for a latent bottleneck after extensive dual-task practice of a visual-manual and an auditory-verbal task. Quarterly Journal of Experimental Psychology.)

Line 68: At this point, it is relevant to review relevant EEG literature on PRP dual tasks. How is previous EEG research investigating PRP dual tasks? What are the investigated task components? What are the main findings of this field? What are the limitations? Please, also refer to Töllner et al. (2012: Töllner, T., Strobach, T., Schubert, T., & Müller, H. (2012). The effect of task order predictability in audio-visual dual task performance: Just a central capacity limitation? Frontiers in Integrative Neuroscience, 6:75.)

In general, I think that the manuscript would strongly benefit by applying the methodological approach of HMP to another data set. So far, the conclusions are exclusively based on a single experimental set of data from 24 participants. This manuscript would benefit a lot, when the current findings could be replicated and specified, and in this way, the manuscript could make a strong impact in the field.

**Have the authors made all data and (if applicable) computational code underlying the findings in their manuscript fully available?**

Reviewer #1: Yes

Reviewer #2: None

PLOS authors have the option to publish the peer review history of their article (what does this mean?). If published, this will include your full peer review and any attached files.

Reviewer #1: No

Reviewer #2: No

**Figure resubmission:**
---

## [Decision Letter · Decision Letter 1]

7 May 2026

Dear Mr. den Otter,

We are pleased to inform you that your manuscript 'Trial-level sequence modeling reveals hidden dynamics of dual-task interference' has been provisionally accepted for publication in PLOS Computational Biology.

Best regards,

Boris S. Gutkin

Academic Editor

PLOS Computational Biology

Andrea E. Martin

Section Editor

PLOS Computational Biology

Reviewer's Responses to Questions

**Comments to the Authors:**

Reviewer #1: The authors have addressed my concerns as much as they can.

Reviewer #2: I have already participated in the first round of peer review. I am convinced by the authors' responses to this first round, and I approve the publication of the manuscript.

**Have the authors made all data and (if applicable) computational code underlying the findings in their manuscript fully available?**

Reviewer #1: Yes

Reviewer #2: None

PLOS authors have the option to publish the peer review history of their article (what does this mean?). If published, this will include your full peer review and any attached files.

Reviewer #1: No

Reviewer #2: No

---

## [Editor Report · Acceptance letter]

PCOMPBIOL-D-25-02374R1

Trial-level sequence modeling reveals hidden dynamics of dual-task interference

Dear Dr den Otter,

I am pleased to inform you that your manuscript has been formally accepted for publication in PLOS Computational Biology. Your manuscript is now with our production department and you will be notified of the publication date in due course.

With kind regards,

Zsofia Freund
